# Microgravimetric Modeling—A New Method for Extracting Adsorption Parameters of Functionalized MIL-101(Cr)

**DOI:** 10.3390/nano13142072

**Published:** 2023-07-14

**Authors:** Xu Zhang, Bo Tian, Zhiheng Ma, He Wang, Zhixuan Cheng, Jiaqiang Xu

**Affiliations:** NEST Laboratory, Department of Chemistry, College of Science, Shanghai University, Shanghai 200444, China; 20720226@shu.edu.cn (X.Z.); tianboskill@163.com (B.T.); mzh139863@163.com (Z.M.); wanghe95@shu.edu.cn (H.W.); zxcheng@shu.edu.cn (Z.C.)

**Keywords:** microgravimetric analysis method, metal−organic frameworks (MOFs), formaldehyde sensor, thermodynamic parameters, kinetic parameters

## Abstract

As a volatile air pollutant, formaldehyde can enter people’s living environment through interior decoration, furniture and paint, causing serious harm to human health. Therefore, it is necessary to develop a sensor for the real-time detection of formaldehyde in low concentrations. According to the chemical interaction between amino groups and formaldehyde, a MIL-101(Cr) aminated-material-based formaldehyde cantilever sensor was prepared, of which ethylenediamine- functionalized MIL-101(Cr) named ED-MIL-101(Cr)) showed the best gas sensing performance. Using quasi-in situ infrared spectroscopy, ED-MIL-101(Cr) was found bound to formaldehyde through a Schiff base. The adsorption enthalpy of formaldehyde-bound ED-MIL-101(Cr) was −52.6 kJ/mol, which corresponds to weak chemical adsorption, so the material showed good selectivity. In addition, ED-MIL-101(Cr) has the most active sites, so its response value to formaldehyde is larger and it takes longer to reach saturation adsorption than bare MIL-101(Cr). Through the research on the gas sensing performance of functionalized MIL-101(Cr) material, we found that it has a strong application potential in the field of formaldehyde monitoring, and the material performance can be quantitatively and accurately evaluated through combining calculation and experimentation for understanding the gas sensing mechanism.

## 1. Introduction

As a harmful volatile organic compound (VOC), formaldehyde will be released into the air from building materials, plywood, paint and other home improvement materials, which will harm the human body and may cause cancer, respiratory diseases, immune system damage and nervous system diseases [1,2,3]. The National Institute of Occupational Safety and Health (NIOSH) and the World Health Organization (WHO) set exposure limits for formaldehyde vapor at 1 ppm and 0.08 ppm, respectively. Therefore, it is very important to achieve rapid and accurate detection of formaldehyde [4,5]. Compared with chromatographs and mass spectrometers that are high performance, large scale and expensive, gas sensors are cheaper and more suitable for field detection [6].

At present, to meet the needs of different applications, gas sensors are rapidly evolving [7]. Among diverse gas sensors, mass-sensitive sensors (microcantilever, quartz crystal microbalance and surface acoustic wave) have attracted the attention of various scientists because of their high precision, low power consumption and high stability [8,9,10,11]. The gas sensor works mainly through the mechanical change (vibration or bending) caused by the adsorption/desorption process of gas molecules on the surface of the sensing layer. Therefore, the preparation of gas-sensitive materials that can capture more gas molecules is the key to achieving excellent gas-sensitive sensor performance. At present, organic polymers, metal oxides and graphene have been developed as gas-sensitive materials to detect various gases [12,13,14,15]. However, most of these materials do not contain pore structures and lack gas adsorption sites, which fails at the rapid detection and identification of gases. So, it is necessary to develop and utilize porous materials to improve the sensitivity of gas detection. Metal–organic frameworks (MOFs) are porous materials for gas storage, separation and sensing [16,17,18,19,20]. The large specific surface area, open metal sites and end functional groups of metal–organic framework materials can be used as adsorption sites of gas molecules to improve the gas sensing performance of materials [21].

In practical application, in order to achieve selective adsorption of the target gas and reduce the influence of interfering gas, MOF materials must be designed and optimized in combination with the adsorption and desorption characteristics of the sensing materials, so that gas and materials can be combined through weak chemical interaction [22]. For example, MIL-101(Cr) porous material was modified through introducing different hydrophilic groups (-SO_3_H, -NH_2_) or hydrophobic groups (-CH_3_, -F) [23]. Among them, aminated MIL-101(Cr) is widely used in catalysis, gas separation and electrochemistry. This not only retains the large specific surface and ordered porous structure of MOF materials but also has unique properties due to the existence of functional groups. There are weak chemical interactions between amino groups and formaldehyde, such as hydrogen bond and Schiff base, so aminated MOFs are a potential formaldehyde-sensing material [24].

According to the classical physico-chemical adsorption theory [25,26], Δ*H^θ^* less than 0 kJ/mol and greater than −40 kJ/mol indicates that there is physical adsorption between gas molecules and materials, while Δ*H^θ^* less than −80 kJ/mol implies strong chemical adsorption. For a qualified sensor, it is most suitable that Δ*H^θ^* is slightly less than −40 kJ/mol. Therefore, adsorption enthalpy can be used as an important parameter to evaluate the sensing performance of materials. In addition, activation energy (*Ea*), as a key kinetic parameter, can be used to quantitatively evaluate the adsorption dynamics of functional materials. The higher the *Ea,* the more difficult it is for gas to react to the material, the slower the adsorption rate and the longer the response time, so the activation energy is the key parameter to evaluate whether the material can be used as a formaldehyde capture or sensing material.

Herein, in order to prove the accuracy of the above methods, MIL-101(Cr) and its aminated materials were prepared and combined with a micro-cantilever platform. Its formaldehyde adsorption and gas sensing performances were studied, and adsorption parameters in the sensing process were extracted to accurately evaluate the materials’ performance.

## 2. Materials and Methods

### 2.1. Material

All the chemicals and reagents are analytical grade and provided by Aladdin Company in Shanghai, including chromium nitrate nonahydrate (Cr(NO_3_)_3_∙9H_2_O, 99.0%), terephthalic acid (H_2_BDC, 99.0%) and 2-amino terephthalate (C_8_H_7_NO_4_, 98.0%) obtained from Maclin. N,N-dimethylformamide (C_3_H_7_NO), sodium hydroxide (NaOH, 96%), sodium acetate trihydrate (C_2_O_5_NaH_9,_ 99.9%), ethanol (C_2_H_6_O, 99.9%), anhydrous toluene (C_6_H_5_CH_3,_ 99.5%) and ethylenediamine (C_4_H_11_N, 99%) were purchased from Shanghai Chemical Reagent Co., Ltd. (Shanghai, China).

### 2.2. Synthesis of MIL-101(Cr) and Aminated MIL-101(Cr)

Synthesis of metal–organic framework (MIL-101(Cr)): Cr(NO_3_)_3_∙9H_2_O and H_2_BDC were dissolved in 25 mL (0.05 mol/L) CH_3_COONa solution according to a molar ratio of 1:1 and stirred for 30 min until the dispersion was uniform. The stirred suspension was transferred into a stainless-steel autoclave with Teflon lining, and a hydrothermal reaction was carried out at 483 K for 8 h. The product obtained from the reaction was washed alternatively with deionized water and ethanol three times and dried in a vacuum at 423 K for 12 h to obtain a blue-green product named MIL-101(Cr).

Synthesis of ED-MIL-101(Cr): Firstly, the as-prepared MIL-101(Cr) was dried in a vacuum oven at 423 K for 24 h with 300 mg (containing about 0.4 mmol of trimer chromium oxide unit), then transferred to a three-neck flask, and 30 mL of anhydrous toluene was added. Then, while stirring, 0.3 g of ethylenediamine (5 mmol) was slowly added to the mixture, and the mixture was heated under reflux at 293 K for 12 h. After cooling to room temperature, the product was collected via filtration, alternately washed with water and ethanol and dried in vacuum at 373 K for 8 h to obtain a yellow-green product, which was named ED-MIL-101(Cr).

Synthesis of NH_2_-MIL-101(Cr): Cr(NO_3_)_3_∙9H_2_O and 2-amino terephthalic acid were dissolved in 15 mL (0.3 mol/L) NaOH solution according to a molar ratio of 1:1 and stirred for 15 min until the dispersion was uniform. The stirred suspension was transferred into a stainless-steel autoclave with Teflon lining, and a hydrothermal reaction was carried out at 423 K for 10 h. The product obtained from the reaction was washed with dimethylformamide to remove the residue of 2-aminoterephthalic acid and dried in a vacuum at 373 K for 8 h to obtain a light green product, named NH_2_-MIL-101(Cr).

### 2.3. Characterization

At room temperature, the crystal structure of the prepared materials was analyzed using a polycrystalline X-ray diffractometer (DX 2700). The morphology and structure of the material were observed via scanning electron microscope (JSM-6700F). In order to prove that the functional group was successfully grafted on the material, a Fourier infrared spectrometer (AVATER370 FT-IR) experiment was adopted. The specific surface area and pore structure of the materials were analyzed via an automatic physical and chemical adsorption analyzer (ASAP2020M+C). The gas response of the coated micro-cantilever chip was tested using High-End MEMS Technology, and the results were recorded and analyzed.

### 2.4. Microcantilever Test System

The microcantilever used in this study has been reported before. The dimensions are 90 μm long, 21 μm wide and 1 μm thick. Firstly, the material is dispersed into absolute ethanol and about 0.1 μL of suspension is loaded on the upper surface of the cantilever end region using a commercial micro-manipulator (manufactured by Eppendorf, model: PatchMan NP2). The process control is assisted via inspection under the microscope (made by Leica, model: DM4000), and it can be clearly seen that the material is covered on the test layer. The coated chip was dried in a vacuum at 50 °C.

Xiamen Haienmai Company intelligent comprehensive physical and chemical property analyzer is used for gas sensitivity test and analysis. The resolution of mass change is 0.5 pg (10^−12^ g). Dynamic gas distribution is performed using a flowmeter, and pure N_2_ is mixed with the gas to be measured and diluted to the target concentration. A four-way valve is used to switch the gas to the pure N_2_ concentration to be measured, so that the target gas enters the sensor test cell. Response and recovery before and after gas switching were recorded using the data acquisition system. In order to avoid signal interference caused by gas flow, the flow rate before and after gas switching should be consistent. When the gas to be measured is captured by the sensitive layer material, it will cause a change of resonance frequency. Using the isothermal concentration gradient curves obtained through testing at different temperatures, the thermodynamic and kinetic parameters can be obtained via calculation [27,28]. Refer to Figure 1.

## 3. Results and Discussion

### 3.1. Component and Structure Characterization of the MIL-101(Cr) and Aminated MIL-101(Cr)

Figure 1a is the image of unfunctionalized material MIL-101(Cr), showing a uniform octahedral morphology with a longitudinal length of about 500 nm and a transverse width of about 450 nm with a rough surface. ED-MIL-101(Cr) obtained via grafting amino groups with ethylenediamine can still maintain the octahedral morphology with a particle size of 300–400 nm [29]. Due to the presence of amino groups in the benzene ring of ligand 2-aminoterephthalic acid, steric hindrance is increased, so the NH_2_-MIL-101(Cr) sample shows uniform particle accumulation, and the particles become smaller, with a particle size of 30~40 nm.

As shown in Figure 2a, the distribution patterns of the three MOFs are in good agreement with the analog signals obtained from the crystal information file. Only the wide Bragg diffraction phenomenon appeared in the aminated material, which indicated that the crystallinity of the material was poor [30,31]. According to the image in Figure 2b, the two peaks at 3470 cm^−1^ and 3340 cm^−1^ in the NH_2_-MIL-101(Cr) spectrum correspond to the asymmetric and symmetric vibration of the amino group (-NH_2_) and the benzene ring of the ligand. Due to the bending vibration absorption of amino groups, the FT-IR spectrum peak at 1660 cm^−1^ appears. It shows that the material has successfully achieved amino functionalization. In the spectrum of ED-MIL-101(Cr), the peaks of 3200–3500 cm^−1^ are the tensile vibration of amino groups in aliphatic group, and the peaks of 2970 cm^−1^ and 1050 cm^−1^ can be indexed to the tensile vibration of C-H and C-N in aliphatic group, respectively, which is different from the characteristics of amino groups in the aromatic group [31,32]. 

From Table 1, the test results of a specific surface area show that MIL-101(Cr) with a large specific surface area was obtained via the hydrothermal method, which provided more gas adsorption sites. After amino functionalization, the material still maintains a large specific surface area and high porosity. Due to the presence of amino groups in the pores, the specific surface areas of both are lower than those of MIL-101(Cr), and the pore volume of ED-MIL-101(Cr) is 0.89 cm^3^/g and the pore diameter is 20.07 Å. However, in 2-amino terephthalic acid, the amino chain is shorter, which has less influence on the pore, so the pore volume of NH2-MIL-101(Cr) is 1.60 cm^3^/g, and the pore size is basically unchanged, which is 19.60 Å. 

### 3.2. Sensing Performance of the MIL-101(Cr) and Aminated MIL-101(Cr)

Through loading the materials on cantilever platform, the performance of the sensors was evaluated at room temperature. Figure 3a shows selectivity for 6 ppm VOC gas, and it shows that aminated materials improve the selectivity to formaldehyde, and there is no significant difference in the response values to interfering gases (benzene series, ethanol, acetone, etc.). ED-MIL-101(Cr) shows the best performance. According to the image in Figure 3b, the 6 ppm formaldehyde response increased from 2.56 Hz for MIL-101(Cr) to 5.26 Hz for ED-MIL-101(Cr) and 4.28 Hz for NH_2_MIL-101(Cr). The reaction and recovery processes of the two materials are different (Figure 3c,d). ED-MIL-101(Cr) can be divided into fast response, slow response, fast recovery and slow recovery and the time is 68 s, 293 s, 54 s and 427 s, respectively. For NH_2_-MIL-101(Cr), there are only two stages, namely fast reaction and fast recovery, which last 56 s and 117 s, respectively. At the same time, although the time in the rapid response phase is shorter, since there are more sites of action of ED-MIL-101(Cr), the response value of ED-MIL-101(Cr) is larger, and the time it takes to reach saturation adsorption becomes longer. In addition, it is also related to the adsorption activation energy of the material. The higher the activation energy, the more difficult the adsorption, the slower the speed and the longer the response time.

Figure 4a,b display the response values of the formaldehyde gas sensor from 0.25 ppm to 10 ppm at room temperature. As the gas concentration increases, the sensor’s response gradually increases. The adsorption curve of the material is fitted, which is very consistent with the Langmuir equation function, as shown in Figure 4c,d. Compared with the two models, the model of ED-MIL-101(Cr) is more satisfied with the Langmuir model, which shows that the adsorption of formaldehyde by low-concentration materials is a reversible chemical adsorption.

## 4. Study on Gas Sensing Mechanism of MIL-101(Cr) and ED-MIL-101(Cr)

### 4.1. Study on Formaldehyde Adsorption Model of Aminated MIL-101(Cr)

In order to study the adsorption models of different aminated MIL-101(Cr) for formaldehyde, quasi-in situ infrared spectroscopy was used in this paper. First, the infrared spectra of ED-MIL-101(Cr) and NH_2_-MIL-101(Cr) without formaldehyde adsorption were tested, and then they were treated with formaldehyde steam for 5 min, and the scanning was performed 64 times. The results are shown in Figure 5.

According to Figure 5a, it can be found that the tensile vibration of -NH_2_ is attenuated at 3200 cm^−1^. The absorption peak at 1625 cm^−1^ corresponds to the bending vibration of -NH-, proving that the -NH_2_ of the materials has converted to -NH- upon absorption of formaldehyde, which is consistent with the infrared change in Schiff’s base reaction. The quasi-in situ infrared spectrum of NH_2_-MIL-101(Cr) after formaldehyde absorption is shown in Figure 5b. It is found that the symmetry of amino groups is reduced according to the tensile vibration of -NH_2_ at 3360 cm^−1^ and 3460 cm^−1^; it may be caused by the hydrogen bond interaction between -NH_2_ and formaldehyde. Therefore, hydrogen bonding between NH_2_-MIL-101(Cr) and formaldehyde is considered. The two aminated materials have different action modes for formaldehyde adsorption, so the adsorption parameters of these two adsorbed processes need to be further investigated.

### 4.2. Calculation of Adsorption Parameters

Temperature-varying experiment for isotherms and adsorption enthalpy (Δ*H^θ^*). According to the mass sensitivity (1 Hz/pg) of the cantilever sensor, the frequency change is converted into adsorption mass. The same frequency change at different temperatures means that the number of molecules captured by the material is the same (the same part covers *θ*). According to the isothermal adsorption curves at different temperatures, we can calculate Δ*H^θ^* using the Clausius–Clapeyron equation with the same coverage. For a qualified sensor, Δ*H^θ^* slightly less than −40 kJ/mol and greater than −80 kJ/mol is the most suitable, meaning it has good adsorption specificity for gas molecules, endowing good repeatability on the sensor, which can realize the long-time cyclic adsorption of gas molecules on the surface of the material. Figure 6 shows that applying the adsorption isotherms of formaldehyde at different temperatures with the above method, the adsorption enthalpies of three materials can be determined: Δ*H^θ^*_MIL-101(Cr)_ = −32.3 kJ/mol, Δ*H^θ^*_ED-MIL-101(Cr)_ = −52.6 kJ/mol, Δ*H^θ^*_NH2-MIL-101(Cr)_ = −46.7 kJ/mol. According to the calculation results, the interaction between MIL-101(Cr) and formaldehyde is physical adsorption, while the role of aminated MIL-101(Cr) is weak chemical interaction, showing that the introduction of an amino group enhances the chemical adsorption of the material and improves the gas detection performance of the material.

Equilibrium Constant (*K^θ^*). According to the fitting curve in Figure 4, the fitting curve of the material conforms to the Langmuir model. It shows that formaldehyde molecules tend to be adsorbed on materials at low pressure, so the fractional coverage *θ* can be expressed as follows:(1)$$\theta =Kp/(1+Kp).\;K=K_{a}+K_{d}$$
where *K^θ^* is the equilibrium constant. According to the literature [33], *θ* can be expressed as *V/V_∞_*_,_ where *V* is the volume of adsorbed formaldehyde and *V_∞_* is the volume of formaldehyde for complete coverage. Equation (1) can be converted to:(2)$$p/V=p/V_{\infty }+{\left(KV_{\infty }\right)}^{-1}$$

When calculating Δ*H^θ^,* we have transformed the frequency change during the test into the mass of formaldehyde adsorbed by the material. According to the molecular weight of formaldehyde (30.0 g/mol), the amount of formaldehyde adsorbed can be obtained. Based on the equation *pV = nRT*, the volume of adsorbed formaldehyde can be obtained, and the images of *p/V* and *p* can be drawn, as shown in Figure 7. According to the image, the data of MIL-101(Cr) can be linearly fitted as *p/V* = 2.51 × 10^9^ *p* + 3.66 × 10^7^. According to Equation (2), it can be concluded that *K* = 1.47 Pa^−1^ and *K^θ^* = *K × p^θ^* = 1.47 × 10^5^. Similarly, the *K^θ^* value of ED-MIL-101(Cr) is 2.10 × 10^5^ and the *K^θ^* value of ED-MIL-101(Cr) is 3.87 × 10^5^. The Langmuir equilibrium constant *K^θ^* reflects the properties of the adsorbents. The larger *K^θ^* is, the better the adsorption performance of the material is.

Adsorption site (*N*). The adsorption of formaldehyde by the two materials is a spontaneous process. Substituting *K* and *p* into Equation (1), a constant fractional coverage *θ* can be obtained, according to *n = Nθ*. Then, the total number of the absorbing sites *N* can be known. The calculated results show that the number of adsorption sites of MIL-101(Cr) is 5.48 × 10^−14^. After amino functionalization, the number of adsorption sites increases, where ED-MIL-101(Cr) is 1.07 × 10^−13^ and NH_2_-MIL-101 is 5.95 × 10^−14^. Due to the different positions of the amino groups, the exposure of the amino groups is affected. There are more sites for ED-MIL-101(Cr) and its response value to formaldehyde is larger.

Gibbs free energy (Δ*G^θ^*). With the obtained *K^θ^* for a certain temperature of *T*, Δ*G^θ^* can be calculated from the relation Δ*G^θ^* = −RT ln*K^θ^*. In chemical thermodynamics, Gibbs free energy is used to assess the direction, and spontaneous reactions always proceed in the direction of the reduction in Gibbs free energy up to equilibrium. Therefore, for the reversible reaction, Δ*G^θ^* < 0 means that the positive reaction occurs spontaneously. The Gibbs free energies of the three materials are Δ*G^θ^*_MIL-101(Cr)_ = −28.8 kJ/mol, Δ*G^θ^*_ED-MIL-101(Cr)_ = −30.4 kJ/mol, Δ*G^θ^*_NH2-MIL-101(Cr)_ = −31.1 kJ/mol., all of which are less than 0, suggesting that their adsorption of formaldehyde at room temperature is a spontaneous process. Furthermore, the Gibbs free energy of the materials decreased after amination, meaning that the introduction of amino groups enhanced the materials’ ability to adsorb formaldehyde.

Entropy change (Δ*S^θ^*). Δ*S^θ^* can be calculated from Δ*G^θ^ =* Δ*H^θ^ − T*Δ*S^θ^*. Entropy change refers to the change in the chaotic degree of the system. When entropy becomes positive, the degree of chaos of the system increases, while when entropy becomes negative, it decreases. After calculation, the entropy of the three materials is Δ*S^θ^*_MIL-101(Cr)_ = −11.7 J/K, Δ*S^θ^*_ED-MIL-101(Cr)_ = −74.5 J/K, Δ*S^θ^*_NH2-MIL-101(Cr)_ = −38.9 J/K, and it can be found that chaos decreases as formaldehyde is trapped by the materials.

Activation energy (*E_a_*). The adsorption process at two temperatures was recorded in real time using a resonant microcantilever. Through solving the Arrhenius formula, the kinetic parameter *E_a_* is extracted. According to the adsorption site, *N* of ED-MIL-101(Cr) and NH_2_-MIL-101(Cr) can be obtained, which can be used for the adsorption equilibrium constant (*k_a_*). In the initial stage of material adsorption, the adsorption rate can be obtained according to the slope of Figure 8. *df/dt = k_a_pN*. Based on the fitting results, the adsorption equilibrium constant of ED-MIL-101(Cr) at 298 K and 313 K was obtained. For comparison, we also calculated the equilibrium constant of NH_2_-MIL-101(Cr), with the associated process reference supplement.
(3)$$ln\frac{k_{a2}}{k_{a1}}=\frac{E_{a}}{R}\;(\frac{1}{T_{1}}-\frac{1}{T_{2}})$$

According to Equation (3), the activation energy of formaldehyde adsorption by ED-MIL-101(Cr) can be calculated to be 14.87 kJ/mol, and that by NH_2_-MIL-101(Cr) can be calculated to be 10.04 kJ/mol. According to the calculation, the faster the adsorption rate of the material, the smaller the activation energy that needs to be overcome. The details of three materials list in Table 2.

According to the calculation results, ED-MIL-101(Cr) has the best adsorption parameters, and its adsorption enthalpy belongs to the weak chemical adsorption region, which is smaller than NH_2_-MIL-101(Cr), so it is easier to react with formaldehyde and shows the best selectivity. And ED-MIL-101(Cr) has the most adsorption sites, so its response value to formaldehyde is the largest. The Δ*G^θ^* < 0 of the three materials shows that their adsorption of formaldehyde is a spontaneous process. Depending on the size of the entropy change, the sum of the three values is less than 0, showing that the degree of disorder of the material after gas absorption decreases and the absolute value of the entropy change is larger because there are more adsorption sites on the surface of ED-MIL-101(Cr). In addition, the number of adsorption sites also affects the adsorption rate. The places of ED-MIL-101(Cr) are the most widespread, so it takes longer to reach saturation adsorption, and the activation energy calculation results can also prove this point. The adsorption activation energy of ED-MIL-101(Cr) is larger, the adsorption rate is slower and the response time is longer, but the difference is not big because there is little difference between the activation energies of the two materials. In summary, ED-MIL-101(Cr) shows the best performance.

## 5. Conclusions

In this paper, MIL-101(Cr) was prepared and aminated, and a resonant gas sensor was constructed through combining it with a micro-cantilever beam, and the response characteristics of the sensing material to formaldehyde gas were systematically studied. It was found that the materials functionalized with amino groups showed good formaldehyde sensing performance. Infrared characterization reveals that there is Schiff-base interaction between ED-MIL-101(Cr) and formaldehyde but hydrogen bond interaction in NH_2_-MIL-101(Cr). Combined with calculation, the adsorption enthalpy in the sensing process is −52.6 kJ/mol and −46.7 kJ/mol for ED-MIL-101(Cr) and NH_2_-MIL-101(Cr), respectively, meaning a satisfactory chemical adsorption and endowing the materials with good selectivity. The activation energy calculation results show that the adsorption activation energy of NH_2_-MIL-101(Cr) is 10.04 kJ/mol, which is lower than that of ED-MIL-101(Cr) (14.87 kJ/mol), and shows a faster adsorption/desorption rate in the gas sensing test, and the response and recovery curve of NH_2_-MIL-101(Cr) is shorter. Based on this method of calculating gas adsorption parameters, researchers can quickly and conveniently study the adsorption model of materials and reveal the sensing mechanism of materials.

## Data Availability

Not applicable.

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
