# Peer review of "Microgravimetric Modeling—A New Method for Extracting Adsorption Parameters of Functionalized MIL-101(Cr)"

_nanomaterials, 2023, doi:10.3390/nano13142072_

Round 1

Reviewer 1 Report

In the manuscript by Xu Zhang et al. a metal–organic framework MIL-101(Cr) and its aminated derivatives were prepared, combined with the micro-cantilever platform, its formaldehyde adsorption and gas sensing performances were studied, and adsorption parameters in the sensing process were extracted. Authors showed that the materials functionalized with amino groups showed good formaldehyde sensing performance because Schiff-base or hydrogen bond interactions leading to a satisfactory chemical adsorption and the materials good selectivity. The manuscript presents interesting results and will be of interest to readers of Nanomaterials, however, after some revision, listed below.

 - The article should be supplemented with the references to the current state of research in the field of metal–organic framework MIL-101(Cr) and its derivatives. The main structural motifs of the metal–organic framework MIL-101(Cr) and its aminated derivatives should be presented in the supplementary figure.

- The article lacks data on porosity, pore size distribution, specific surface area of the synthesized metal–organic framework MIL-101(Cr) and its derivatives. The article should be supplemented with these data. It is important to show the change in these characteristics for functionalized derivatives.

- It is not clear to me why the authors did not achieve a saturation of the sensor response (Fig.3, 4) for the ED-MIL-101 (Cr) sample. What will happen with a longer exposure time?

- In order to talk about good sensory properties, it is necessary to analyze the stability of the signal and the stability of the baseline in long-term experiments. The presented data (Fig. 4) contain only 4 cycles at increasing analyte concentration. What happens when concentration decreases? Authors should complete the study of sensory properties in long-term experiments.

- The authors demonstrated a good response by increasing adsorption because Schiff-base or hydrogen bond interactions. At the same time, sensor properties imply dynamic behavior. Does desorption process proceed well enough under the chosen conditions?

- Interfering gases (benzene series, ethanol, acetone) were used for selectivity analysis. However, what will be the response for other aldehydes and its derivatives, such as acetaldehyde?

Reviewer 2 Report

see attached file

The paper is written in well-understandable English. However, there are numerous small grammatical errors which can easily be detected and corrected by a native speaker. Occasionally, there are also strange sentences that require correction but also some guess-work on the side of the person doing the amendments.  

Round 2

Reviewer 1 Report

I have reviewed the revised manuscript. The authors took into account all my comments. I recommend the manuscript for publication.

Reviewer 2 Report

OK with me. Accept.

Minor amendments by native speakers.